# Topic Jerk Detector: Detection of Tweet Bursts Related to the Fukushima Daiichi Nuclear Disaster

**Hiroshi Nagaya [1,*], Teruaki Hayashi [1], Hiroyuki A. Torii [2] and Yukio Ohsawa [1]**

[1] School of Engineering, The University of Tokyo, Tokyo 113-8656, Japan; hayashi@sys.t.u-tokyo.ac.jp (T.H.); ohsawa@sys.t.u-tokyo.ac.jp (Y.O.)

[2] School of Science, The University of Tokyo, Tokyo 113-0033, Japan; torii@chem.s.u-tokyo.ac.jp

* Correspondence: hiroshi-nagaya@g.ecc.u-tokyo.ac.jp

**Abstract:** In recent disaster situations, social media platforms, such as Twitter, played a major role in information sharing and widespread communication. These situations require efficient information sharing; therefore, it is important to understand the trends in popular topics and the underlying dynamics of information flow on social media better. Developing new methods to help us in these situations, and testing their effectiveness so that they can be used in future disasters is an important research problem. In this study, we proposed a new model, "topic jerk detector." This model is ideal for identifying topic bursts. The main advantage of this method is that it is better fitted to sudden bursts, and accurately detects the timing of the bursts of topics compared to the existing method, topic dynamics. Our model helps capture important topics that have rapidly risen to the top of the agenda in respect of time in the study of specific social issues. It is also useful to track the transition of topics more effectively and to monitor tweets related to specific events, such as disasters. We attempted three experiments that verified its effectiveness. First, we presented a case study applied to the tweet dataset related to the Fukushima disaster to show the outcomes of the proposed method. Next, we performed a comparison experiment with the existing method. We showed that the proposed method is better fitted to sudden burst accurately detects the timing of the bursts of the topic. Finally, we received expert feedback on the validity of the results and the practicality of the methodology.

**Keywords:** social media; text mining; burst detection; crisis situation

## 1. Introduction

This paper is an extended version of our previous conference paper [1]. On 11 March 2011, catastrophic accidents took place at the Fukushima Daiichi nuclear power plant. These accidents resulted in widespread radioactive contamination and radiation exposure [2]. Residents in the surrounding area were exposed to radiation over a long period, and the fear of spreading contamination has caused social unrest throughout Japan [3]. Accurate and fast information sharing is considered essential for survival in such situations. However, during disaster situations, information collection is challenging, because of traffic congestion and extensive damage to the network infrastructure, which prevents the press from assessing the situation of the affected areas [4].

In disaster situations, social media is very informative and useful for timely information sharing, owing to its unique features. Social media allows the rapid dissemination of information to a wide range of population and allows real-time communication [5]. Some studies have shown that social media is reliable and provides valuable informational content and has a very high potential to be a useful tool in situations where a quick emergency response is essential [6]. Twitter, one of the most famous social media platforms, allows users to distribute short messages ("tweets") using the

World Wide Web or smartphone apps [7]. Twitter users can share other people's tweets with their friends, in a process known as "retweeting," which is the critical mechanism of information diffusion on Twitter [8]. The number of Twitter users reached 200 million in March 2011 [9]. To develop effective information dissemination methods in future crises, it is important to analyze the log data of social media efficiently with such backgrounds. It is also important to grasp the occurrence of events using social media, and identify ways to propagate the necessary information. For proper information dissemination and management, we need to be aware of several phenomena in social media. For instance, when we want to prevent flaming phenomena, it is necessary to analyze and monitor the posted contents to identify the cause of these phenomena [10]. To do so, it is useful to understand what topics strongly attract the users' attention with respect to time, especially at the moment of their occurrence. Therefore, we have developed a model that accurately captures the moment when the topics strongly attract the users' attention ("bursting"). In this study, we design a method for detecting the dynamics of the streams of topics on Twitter, and extracting "hot" topics at every point after the crisis-associated event. The proposed method, "topic jerk detector," is more effective for detecting bursting than an existing method, especially in terms of figuring out the specific topics the public has a great deal of interest in. Due to this feature, it is expected that the proposed method can recognize hot words in each term among users and track the transition of topics more effectively. Such a feature makes our method useful for monitoring tweets related to specific events, such as disasters and will be useful as a reference during the formulation of information sharing policy.

## 2. Related Works

In this section, we introduce some research related to this study, primarily from two perspectives, tweet analysis for Fukushima Daiichi nuclear disaster and modeling streams of topics and burst detection. We also remark on the contribution of this study to existing research.

### 2.1. Tweet Analysis for Fukushima Daiichi Nuclear Disaster

Analysis of crisis-related Twitter data is an important research topic, and several researchers have actively investigated it [11–14]. Some researchers analyzed Twitter data related to the Fukushima Daiichi nuclear disaster; this disaster is one of the most significant recent events in which social media was actively used for information dissemination. The existing approaches can be approximately divided into two categories. The first category is focused on the interactions of users, e.g., direct communication and information sharing at the time of the disaster. For example, Toriumi et al. analyzed the tweets that were posted before and after the accident, to unravel how people share disaster-related information on Twitter. These researchers studied replies and retweets based on whether interactions occurred in the network of the users' followers [4]. They concluded that social media users changed their behavior to widely disseminate important information, and shared fewer non-emergency tweets to avoid interfering with critical disaster-related information. Sakaki et al. analyzed user interactions with regional information [15]. They concluded that the users diffused substantially more information after the earthquake, particularly in the heavily damaged areas; the users communicated with nearby users but diffused information posted by distant users.

Some studies were focused on the transmission of information regarding the Fukushima Daiichi nuclear disaster, using social media. Rantasila et al. examined how the fifth anniversary of the Fukushima Daiichi nuclear disaster was commemorated by the English-speaking community of Twitter users and included "#fukushima" in March 2016 [16]. They found that while most of the (influencing) retweets contained the ritual elements of collective memory, grief, and observance, another prominent feature was a strongly politicized discourse surrounding the aftermath of the disaster. Aoki et al. analyzed approximately 19 million tweets related to radiation that had been posted for the duration of one year after the Fukushima Daiichi nuclear disaster and revealed how the number of tweets by citizens containing radiation-related information had evolved regionally [17]. They concluded that the number of tweets per 100,000 users became half of that in March 2011 after 3 or 4 months from the

Fukushima Daiichi Nuclear Plant disaster, for three districts except the Fukushima Prefecture district; half a year later, that number became half in Fukushima Prefecture. Tsubokura et al. analyzed tweets that were poseted within half a year after the accident. The analysis found that many of the retweets about the Fukushima disaster were based on original posts by only a few hundreds of influencers [18]. Using a clustering algorithm, they found three influencers' group and characterized them in terms of the contents of their tweets. Although essential from the viewpoint of appropriate information transmission and prevention of misinformation, the number of studies of the second type (that analyze the dissemination of "hot" topics throughout the community of users, particularly over long periods post-crisis) was limited. Thus, in this study, we analyzed tweets related to the Fukushima Daiichi nuclear disaster that was posted over six years after the accident.

### 2.2. Modeling Streams of Topics and Burst Detection

It is useful to model streams of topics and detect the bursting ("hot") topics for each period, to monitor the occurrence of topics in a stream of events. Some research was conducted on these methodologies, and we review some representative ones. Kleinberg's burst model [19] is a groundbreaking method that presents an intuitively appealing definition of the word "burst." Kleinberg defined burst as an increase in the arrival rate of a term in a stream of text, and develops an automaton for tracking an optimized estimate of this rate. Initially motivated to model the phenomenon of gamma-ray bursts, Shasha developed several methods for identifying bursts [20,21] based on the hierarchies of fixed-length time intervals. Some of the limitations of these methods are the expensive computational cost and vague definitions of burst strength. Many recently developed machine learning approaches are characterized by the same problems [22,23]. In an attempt to solve these problems, "topic dynamics" was proposed as an alternative model [24]. This framework uses the physical intuition, modeling bursts as intervals of increasing momentum; it can be applied to several "trend" quantities of interest, such as changes in stock values or changes in the impact of a scientific article. Therefore, topic dynamics is very suitable for situations, such as crises, where rapid data-based information sharing services are required [25,26].

In this paper, "topic" for each period is defined as the most representative keywords mentioned by the users. In methods such as "topic model" [27–29], it is superior in that the topic can be taken as a semantic-level description of the text. The topic models are hierarchical probabilistic models to find patterns of words in document collections [30]. One of the most representative topic models is latent Dirichlet allocation (LDA) [31]. The method is a three-level hierarchical Bayesian model that models each item in the collection as a finite mixture over an underlying set of topics. It also models each topic as an infinite mixture over an underlying set of topic probabilities. Several studies apply topic models such as LDA for Twitter data [32–34]. For example, Zhao et al. applied a topic model, LDA to Twitter contents and compared them with those of the New York Times [35]. In this study, we focus on the topic dynamics for the reasons above, taking into account its immediacy and the convenience of using fewer resources. Furthermore, since Twitter expressions are sometimes ambiguous, we do not automate the extraction of a semantic-level description of the text in this study. Only the top keywords are provided as material to be interpreted by humans. However, comprehensive analysis with topic model methods will deepen our understanding of events.

Therefore, we focus on topic dynamics for the task of analyzing tweets related to disaster situations, and examine some cases where it was applied. Tamura et al. developed a method for identifying local temporal burstiness to detect local hot keywords based on the users' location [36]. Xie et al. applied the core idea of topic dynamics to some tasks such as real-time bursty topic detection on Twitter [37]. In this study, we developed a method termed "topic jerk detector," an expansion of topic dynamics; the topic jerk detector is more effective for detecting "bursting" than topic dynamics, especially in terms of figuring out the specific topics the public has a great deal of interest in. The model is a more abrupt-bursting-fit and ideal for identifying topic bursts, compared to the existing method, topic dynamics. This feature helps recognize the hot words in each period among users by providing

the moment of bursting for each main topic with respect to time. This makes it possible to monitor the occurrence of topics in a stream of events, especially topics that interest the public immensely.

## 3. Dataset

### *3.1. Data Extraction*

In this study, we employed the same dataset used in a previous study [18], and the key phrases used as the extraction conditions are shown in Table A1. The key phrase filter was used to extract Japanese tweets related to the Fukushima Daiichi nuclear power plant accidents from all the tweets on the Twitter platform. These key phrases were selected in collaboration with domain experts, and we aimed to collect tweets about radiation and radioactivity in relation to the Fukushima nuclear accident as comprehensively as possible. The dataset consists of all the tweets posted from 1 January 2011, to 30 June 2017. Owing to the large number of targeted tweets and budgetary constraints, we used a sample composed of 8% of the selected tweets. Each tweet analyzed contained at least one of the keywords listed in the table. This sub-sample data was purchased from NTT DATA, a Japanese multinational system integration company. In Figure 1, we show the number of tweets per month. The blue bars indicate the number of original tweets related to the Fukushima disaster, and the red bars indicate the number of retweets. As can be observed, there are still various tweets related to the disaster, even six years after the disaster, and the indicated number of retweets covers the entire sample of tweets. Furthermore, we observe that there are some influential tweets, and the retweeting behavior remains unchanged long after the event.

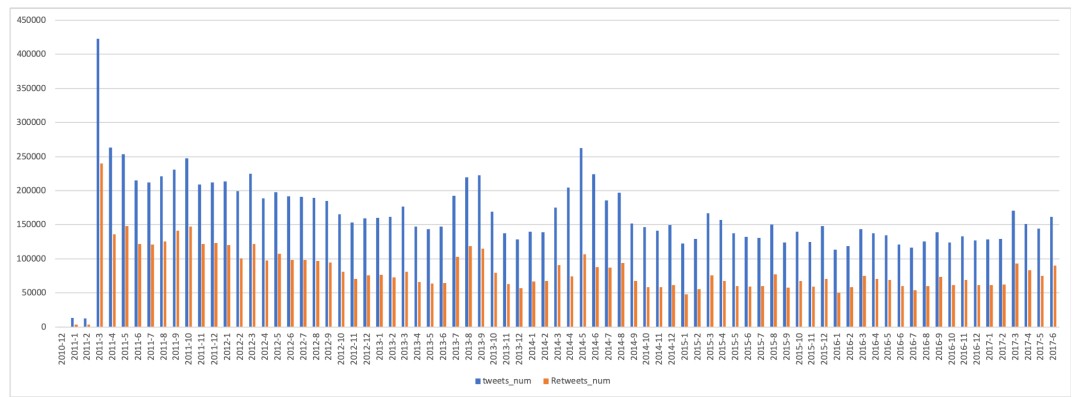

**Figure 1.** Number of original tweets and retweets per month.

### *3.2. Preprocessing*

The dataset included some tweets in which the key phrases were used in a different sense than what was intended. For example, we chose the word "monitoring," which refers to the measurement of radiation levels, as one of the key phrases. Meanwhile, there are some tweets containing the same word in reference to the title of a famous Japanese TV show. To solve this issue, we re-extracted the tweets from the original dataset, and requested that they contain the word "Fukushima," in addition to the prescribed keywords. This change resulted in a new dataset consisting of 21,898,729 tweets and retweets from 2,809,329 users. Subsequently, we used MeCab [38], developed by NAIST, Japan, and NEologd [39] to conduct a morphological analysis and capture the dataset as Bag-of-Words. We removed the uniform resource locators (URLs) and some symbols such as "@[userID]:," "RT @[userID]:," "# (hashtags) ." Several stop words unrelated to the contents such as "they", "I", and "we" were removed; we also used only nouns in this study.

## 4. Method

In this section, we describe "topic dynamics," an existing method for detecting topic bursts. We expand the method based on a physics analogy, and propose the "topic jerk detector," a more effective method for detecting "bursting," compared to the existing method.

### 4.1. Topic Dynamics

Topic dynamics uses the move and change delete (MACD) histogram, which is primarily used to analyze the movement of stock prices in financial markets. In the beginning, we defined $s_{w,t}$ for each word $w$ at time $t$, as shown in Equation (1). This score is equivalent to the price of stocks in financial analysis.

$$s_{w,t} = \sum_{d \in D} \log(\mathrm{RT}_{d,t}) \times \mathrm{TF}_{w,d,t} \times \log\left(\frac{N_t}{\mathrm{DF}_{w,t}}\right) \tag{1}$$

In this equation, $\log(\mathrm{RT}_{d,t})$ expresses the influence of individual tweets. $\mathrm{RT}_{d,t}$ denotes the number of times the tweet $d$ was retweeted at time $t$. $D$ is a set of tweets containing the word $w$. The score computed by $\mathrm{TF}_{w,d,t} \times \log(\frac{N_t}{\mathrm{DF}_{w,t}})$ expresses the general definition of the term frequency-inverse document frequency (TF-IDF) [40] that determines how favorably the word $w$ may be used in the tweet corpus. $\mathrm{TF}_{w,d,t}$ represents the frequency with which the word $w$ appeared in the tweet $d$ at time $t$. $N_t$ represents the total number of related tweets posted by users before time $t$. $\mathrm{DF}_{w,t}$ represents the total number of tweets containing the keyword before time $t$. For the subsequent calculation, we calculated the exponential moving average (EMA) as follows:

$$\mathrm{EMA}^n[s_{w,t}] = \beta \times s_{w,t} + (1 - \beta) \times \mathrm{EMA}_w^{n-1}[s_{w,t}] = \sum_{k=0}^{n} \beta(1 - \beta)^k s_{w,t-k} \tag{2}$$

$$\beta = 2 \times (n + 1)^{-1} \tag{3}$$

The MACD was calculated as the difference between $\mathrm{EMA}^{n_1}[s_{w,t}]$ and $\mathrm{EMA}^{n_2}[s_{w,t}]$ for different time periods $n_1, n_2$ ($n_2 > n_1$):

$$\mathrm{MACD}_w^{(n_1,n_2)} = \mathrm{EMA}^{n_1}[s_{w,t}] - \mathrm{EMA}^{n_2}[s_{w,t}] \tag{4}$$

Finally, the score yielded by topic dynamics is calculated as the MACD histogram, a differential between the MACD and its moving average line (EMA) for periods $n_3$:

$$\mathrm{MACDhistgram}_w^{(n_1,n_2,n_3)} = \mathrm{MACD}_w^{(n_1,n_2)} - \mathrm{EMA}^{n_3}[\mathrm{MACD}_w^{(n_1,n_2)}] \tag{5}$$

Here, the MACD can be interpreted as a first-order differential of $s_{w,t}$, and the MACD histogram (topic dynamics) corresponds to a second-order differential in terms of signal processing [41].

### 4.2. Topic Jerk Detector: Expanded Method for Bursting Topics Detection

Using an analogy from physics, the MACD can be interpreted as "velocity" (first-order differential), and topic dynamics can be interpreted as "acceleration" (second-order differential). As described in Section 2.2, modeling bursts as intervals of increased momentum that can be applied to several "trend" quantities of interest corresponds to physical intuition. Topic dynamics detects word bursts when the MACD histogram value is positive. We propose an extended form of topic dynamics, "topic jerk detector" as follows:

$$\mathrm{TopicJerkDetector}_w^{(n_1,n_2,n_3,n_4)} = \mathrm{MACDhistgram}_w^{(n_1,n_2,n_3)} - \mathrm{EMA}^{n_4}[\mathrm{MACDhistgram}_w^{(n_1,n_2,n_3)}] \tag{6}$$

Furthermore, using another physics analogy, the scores calculated using the topic jerk detector can be regarded as a "jerk" (third-order differential, the derivative of acceleration). A jerk is a physical

quantity that shows a large value in response to a sudden increase in acceleration (for example, when the car starts suddenly, the value of jerk is large). Based on these characteristics, the proposed topic jerk detector is expected to be effective for identifying abrupt bursts. Topic jerk detectors can ident1ify the burst of specific topics on Twitter and the resulting surge of user interest in it. In Table 1, we summarize the methods and their corresponding physical quantities.

**Table 1.** The correspondence between the methods and physical qualities.

| Methods | Physical Quantities Corresponding to the Methods |
|---------|--------------------------------------------------|
| Topic Jerk Detector | Jerk |
| Topic Dynamics | Acceleration |
| MACD | Velocity |

## 5. Experiment

### 5.1. Detection of Hot Topics and Transition Plotting

In this experiment, we applied the topic jerk detector to the dataset presented in Section 3 to detect the topics of interest to users in each period and their temporal variation. Here, we set the unit of period to one week. First, the top 100 retweeted tweets were extracted as "influential tweets" for each week, to reduce computational complexity. As mentioned in Section 2.1, Tsubokura et al. revealed that many of the tweets related to radiation and radioactivity in connection to the Fukushima disaster were retweets of tweets posted by a few hundred accounts (influencers) [18]. Next, we calculated the score for each topic using the topic jerk detector. In each period, we identified the topics that users were interested in by selecting the words with the highest calculated values. Then, it was necessary to determine four parameters $(n_1, n_2, n_3, n_4)$ in Equations (4)–(6). To obtain the values of $(n_1, n_2, n_3)$, we applied the values commonly used in the technical analysis of financial markets $(9, 12, 26)$, and qualitatively set 3 for $n_4$ in this experiment. The result that we obtained from applying the topic jerk detector to our dataset is shown in Tables 2–4. We succeeded in detecting some topics that attracted the users' interest in each period. Tables 2 and 3 shows that the top 20 hot topics whose scores, as calculated using the method, was higher, compared to others, with the detected timing for each year. In the tables, we can see some specific persons whose remarks were picked up on Twitter, and brought to users' attention. In Table 4, we also show, the top 10 pairs of words and timings according to the proposed method had high scores.

**Table 2.** Top 20 popular topics for which the scores calculated using the proposed method were high for each year, from 2011 to 2014.

| Week | Word (2011) | Week | Word (2012) | Week | Word (2013) | Week | Word (2014) |
|------|-------------|------|-------------|------|-------------|------|-------------|
| 2011-36 | minister | 2012-01 | cesium | 2013-29 | Taro Yamamoto | 2014-20 | nosebleed |
| 2011-36 | Hachiro | 2012-32 | Hiroshima | 2013-29 | radioactive waste | 2014-19 | nosebleed |
| 2011-37 | Hachiro | 2012-01 | descent | 2013-36 | Tokyo | 2014-20 | Oishinbo |
| 2011-36 | reporter | 2012-30 | strontium | 2013-31 | spill | 2014-19 | Oishinbo |
| 2011-35 | pain | 2012-37 | thyroid Cancer | 2013-36 | Olympic Games | 2014-18 | nosebleed |
| 2011-36 | Fukushima | 2012-06 | earthworm | 2013-07 | thyroid cancer | 2014-11 | News station |
| 2011-41 | Setagaya | 2012-29 | subcontract | 2013-29 | projected to win | 2014-38 | traffic |
| 2011-15 | Chernobyl | 2012-09 | son | 2013-29 | vegetables | 2014-06 | Tamogami |
| 2011-38 | fireworks | 2012-20 | evacuation | 2013-31 | contaminated water | 2014-45 | photographer |
| 2011-44 | fission reaction | 2012-49 | human | 2013-32 | outflow | 2014-29 | removal |
| 2011-11 | discrimination | 2012-34 | greenling | 2013-44 | Taro Yamamoto | 2014-08 | tank |
| 2011-11 | Fukushima | 2012-14 | wood waste | 2013-30 | Taro Yamamoto | 2014-45 | foreigner |
| 2011-28 | beef | 2012-04 | lump | 2013-32 | contaminated water | 2014-06 | Utsunomiya |
| 2011-40 | thyroid | 2012-02 | cesium | 2013-30 | contaminated water | 2014-15 | Obokata |
| 2011-21 | sv | 2012-43 | kg | 2013-38 | block | 2014-52 | thyroid cancer |

**Table 2.** *Cont.*

| Week | Word (2011) | Week | Word (2012) | Week | Word (2013) | Week | Word (2014) |
|------|-------------|------|-------------|------|-------------|------|-------------|
| 2011-35 | Hosono | 2012-32 | Nagasaki | 2013-30 | steam | 2014-18 | Oishinbo |
| 2011-44 | xenon | 2012-22 | fertilizer | 2013-02 | element | 2014-11 | thyroid cancer |
| 2011-35 | radioactive waste | 2012-01 | Fukushima | 2013-13 | Sazae-san | 2014-29 | scattering |
| 2011-17 | msv | 2012-01 | highest | 2013-15 | contaminated water | 2014-43 | dismantling |
| 2011-39 | plutonium | 2012-26 | plutonium | 2013-29 | interview | 2014-11 | Housute |

**Table 3.** Top 20 popular topics for which the scores calculated using the proposed method were high for each year, from 2015 to 2017.

| Week | Word (2015) | Week | Word (2016) | Week | Word (2017) |
|------|-------------|------|-------------|------|-------------|
| 2015-37 | outflow | 2016-46 | bullying | 2017-14 | voluntary evacuation |
| 2015-17 | drone | 2016-06 | high school students | 2017-18 | fire |
| 2015-43 | leukemia | 2016-42 | subcommittee | 2017-14 | self-responsibility |
| 2015-37 | heavy rain | 2016-38 | dam | 2017-14 | Imamura |
| 2015-09 | contaminated water | 2016-44 | car wash | 2017-08 | lecturer |
| 2015-09 | outflow | 2016-06 | external exposure | 2017-19 | fire |
| 2015-32 | Paris | 2016-42 | resignation | 2017-14 | minister for reconstruction |
| 2015-09 | open sea | 2016-42 | doubt | 2017-08 | foreigner |
| 2015-37 | rainwater | 2016-35 | burden | 2017-18 | forest fire |
| 2015-43 | industrial accident | 2016-07 | thyroid Cancer | 2017-18 | forest |
| 2015-35 | fir | 2016-06 | writing | 2017-08 | Kansai Gakuin University |
| 2015-40 | Naomi Kawashima | 2016-52 | thyroid cancer | 2017-08 | discriminatory remarks |
| 2015-43 | Work Accident Certification | 2016-06 | papers | 2017-08 | Fukushima native |
| 2015-48 | papers | 2016-48 | homeroom teacher | 2017-19 | mountain forest |
| 2015-37 | sandbag | 2016-23 | thyroid cancer | 2017-08 | female student |
| 2015-36 | thyroid cancer | 2016-35 | typhoon | 2017-19 | wildfire |
| 2015-43 | certification | 2016-35 | imputation | 2017-08 | student |
| 2015-41 | Toshihide Tsuda | 2016-52 | reduction | 2017-14 | minister |
| 2015-12 | Ai Otsuka | 2016-16 | cost | 2017-19 | Namie |
| 2015-06 | newspaper | 2016-38 | groundwater | 2017-15 | voluntary evacuation |

**Table 4.** Top 10 pairs of words and timings, for which the scores calculated using the proposed method were high.

| Week | Word (English) | Word (Japanese) | Score |
|------|----------------|-----------------|-------|
| 2017-14 | voluntary evacuation | 自主避難 | 46.202 |
| 2013-29 | Taro Yamamoto | 山本太郎 | 34.372 |
| 2017-18 | fire | 火災 | 33.838 |
| 2017-14 | self-responsibility | 自己責任 | 29.074 |
| 2013-29 | radioactive waste | 放射性廃棄物 | 26.248 |
| 2015-37 | flowing out | 流出 | 25.327 |
| 2017-14 | Imamura | 今村 | 24.445 |
| 2014-20 | nosebleed | 鼻血 | 22.384 |
| 2017-08 | lecturer | 講師 | 22.046 |
| 2014-19 | nosebleed | 鼻血 | 21.96 |

Next, we plotted the scores calculated using the topic jerk detector for some topics in Figures 2 and 3. These show the transition of the level of interest from users. Figure 2 is a graph that plots the scores of the top-9 words picked up in Table 4, and Figure 3 is a graph showing some key-persons that appear in Tables 2 and 3 (Obviously inappropriate ones are left out). From these graphs, we can see that some topics re-emerge after the first burst, and some do not.

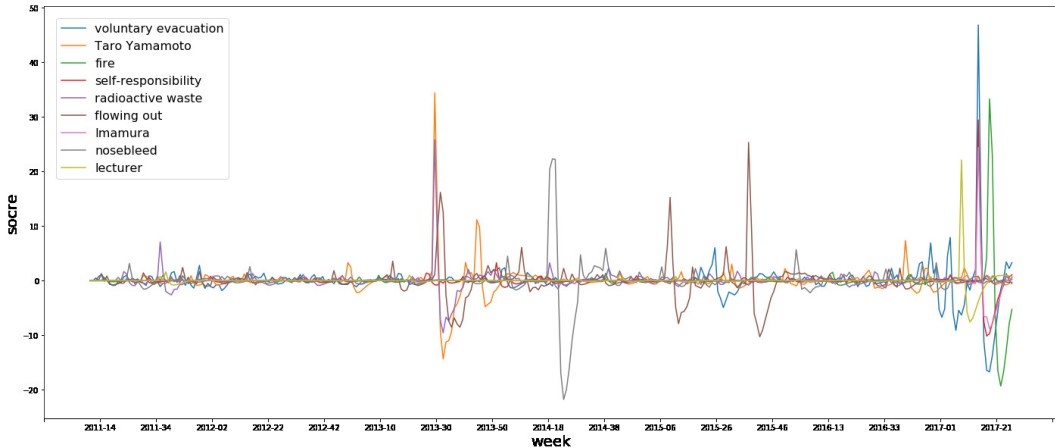

**Figure 2.** Plot of the scores of the top 9 words picked up in Table 4.

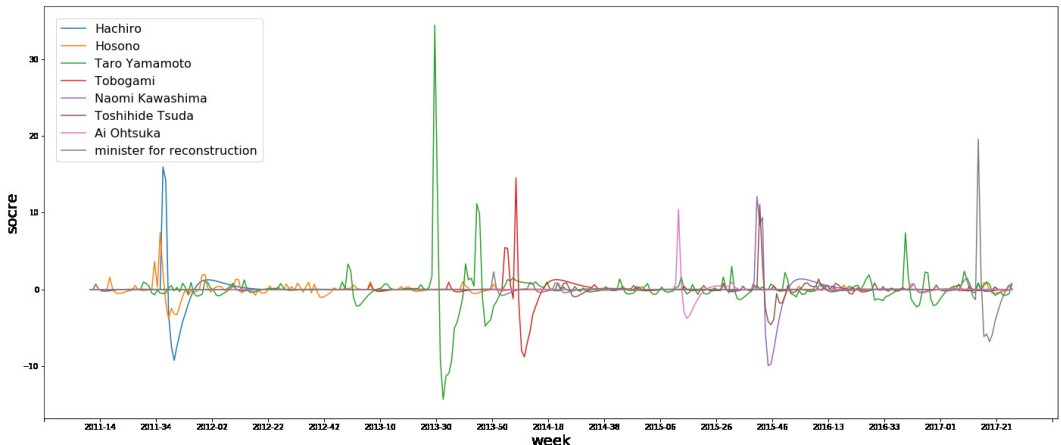

**Figure 3.** Plot of the scores of some key-persons that appear in Tables 2 and 3.

### 5.2. Model Comparison

Our proposed model, the topic jerk detector, detects an outburst on Twitter and a sudden increase in interest in particular topics from users. This experiment verifies that the topic jerk detector, with its unique behavior, is more effective in tracking the rapid upward movement of a topic than the existing method, topic dynamics. We demonstrate that the topic jerk detector is a more abrupt-bursting-fit detector by making comparisons with topic dynamics. We picked up some topics from Table 4, and plotted the values calculated by each method and the number of tweets that include the words around the burst timing in Figures 4 and 5. We set the unit of period to one week, just as in Section 5.1. The green curve shows the values outputted by the topic jerk detector, the blue ones correspond to topic dynamics, and the black bars are the number of tweets for each period. Based on this result, we focused on some parts of these figures, and named them Part1 and Part2. As shown in Section 6, a detailed discussion on the behavior of both models in these parts, it can be said that the topic jerk detector detects the timing of the bursts more accurately, and the topic dynamics method is more effective at capturing continuous trends.

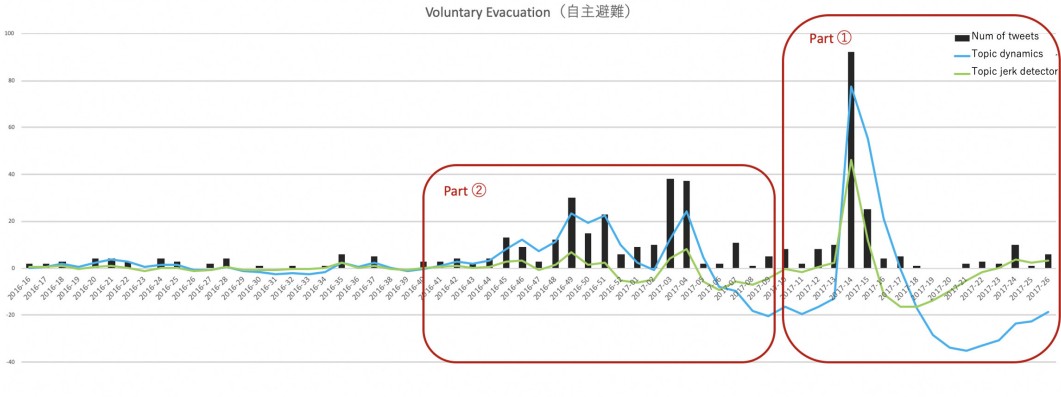

**Figure 4.** Comparison of models to show the topic jerk detector is a more abrupt-bursting-fit extension of topic dynamics in the case of "Voluntary evacuation". The green curve shows the values calculated by the topic jerk detector, the blue ones correspond to topic dynamics, and the black bars are the number of tweets for each week.

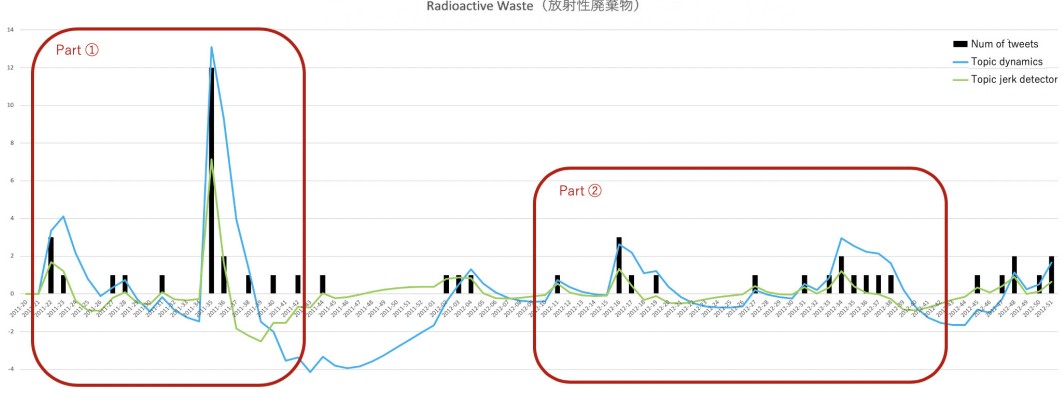

**Figure 5.** Comparison of models to show the topic jerk detector is a more abrupt-bursting-fit extension of topic dynamics in the case of "Radioactive waste." The green curve shows the values calculated by the topic jerk detector, the blue ones correspond to topic dynamics, and the black bars are the number of tweets for each week.

*5.3. Domain Expert Feedback*

To validate our method from other aspects, we interviewed two domain experts and asked them to provide feedback about the method and its performance. The two experts specialized in the fields of immunology and medicine, and they have researched information diffusion related to radiation and radioactivity in connection to the Fukushima disaster. At the time of the interview, we shown Tables 3 and 4 to the experts and asked the following three specific questions.

- Are the results considered appropriate as information diffused to the public for each period in the tables?
- Is there something that says "It is strange that this word has not appeared during this period"?
- Please share the findings obtained during the research and investigation of nuclear power plant accidents related to the analysis results and related matters.

Here, we show the answers from the two domain experts.

**Expert A**

The proposed method succeeded to select topics based on the situation at the time. First, the expert thought that there were several unrelated words, but by comparing them with other words, the expert could understand what they were referring to. For example, when the expert only looked at the word "fire," it did not make sense, but when this word was accompanied by the words "Namie" and/or "radiation splash," the expert was able to recall a series of incidents. Experts can understand these results in detail; however, it is difficult for ordinary people with less knowledge to understand. Although hints can be provided, it is difficult to identify related events using only individual words.

**Expert B**

It is just an enumeration, and it does not answer the question of "how the topic changed." As a person who wants to control information and control flaming phenomena, it is essential to say, in practical terms, how the topic has changed and at what pace. It is essential to answer the question, "How do you explain the results to the public?" This is a severe problem, and there is room for improvement. In the future, it will be necessary to propose and apply a method that focuses on which of the general problems that are highlighted on social media to focus on, and what can be clarified.

## 6. Discussion

From Tables 3 and 4, it can be said that the proposed topic jerk detector succeeded in selecting topics for each period, as Expert A said in Section 5.3. Specifically, there were some tweets to which considerable attention was paid, which included criticism of the government and detailed information on the news, with some URLs to news sites. In this ranking, "Taro Yamamoto" and "Imamura" are the individuals' names of politicians who were strongly associated with the Fukushima disaster. This reflects the tendency, such as in flaming phenomena, that an individual's remarks and behavior are focused on social media. The development of an interface that helps to understand this result in collaboration with our method can be considered to be an essential step for the application of our proposed method, as mentioned by Expert B in Section 5.3. In Figures 2 and 3, we succeeded in plotting the scores as calculated by the topic jerk detector for some topics, and show the transition of the level of interest from users well. It is anticipated that our method will be very useful for monitoring tweets related to specific events, such as disasters, and will be useful as a reference, during the formulation of information sharing policy.

In Part 1 of Figures 4 and 5, as can be observed, these methods simultaneously capture the timing of word bursts. Subsequently, the value yielded by the topic jerk detector drops sharply, and converges to zero faster than that of the topic dynamics method. Thus, based on the results of this study, it may be concluded that the topic jerk detector accurately detects the timing of the bursts, compared to the existing method. This result supports the claim that the proposed method is more effective in monitoring hot topics by performing an accurate periodic detection of the burst timing. On the other hand, looking at Part 2, the occurrences of significant bursts are few, whereas the trend is long-lasting, indicating that the topic dynamics method is more suitable than the topic jerk detector. It can be said that the topic dynamics method is superior at capturing continuous trends. Additionally, a simple frequency analysis of keywords that are shown as black bars in Figures 4 and 5 is sometimes a good way to grasp the topics. It is intuitive and suitable for estimating users' interest simply. Therefore, these techniques should be used differently depending on the purpose.

In Section 5.3, we concluded that our proposed method is effective, and can help to reveal popular topics that attract the attention of social media users. Indeed, based on the top topics presented by our method point by period, such as "Namie" and "radiation splash," Expert A associated the incident "fire," which is an example showing that we can see the possibility of an expert interpreting a series of events by linking the relevant topics at each point in time. It can be said that our method makes it possible to monitor the occurrence of topics in a stream of events, especially topics the public has a great deal of interest in. However, there is still some room for improvement. After the bursting

words were discovered for each period, there is no explanation on how the topic changed, as noted by Expert B. Therefore, we should improve our method and make it more explanatory of the dynamics of topic changes in future. Furthermore, it can be improved to provide recommendations regarding information transitions more easily, which will hopefully help to disseminate important information to the general public efficiently.

It may be meaningful to categorize the bursting topics that are detected using our method into "the topics that fizzled out after bursting" and "the topics that survived and continued to attract the users' attraction after bursting." By doing this, we can expect to gain insights on information transitions/trends; this can facilitate the effective distribution of useful and important information to social media users. There is a need to continue to survey and examine the type of incidents behind specific topic bursts. Furthermore, our method is expected to be further extended. For example, it might be possible to discover hidden topics by revising the definition of Equation (1). It is an important research theme that makes our method more beneficial.

## 7. Conclusions

In disaster situations, it is essential to develop analysis methods to grasp the kind of topics that attract users' interest and to understand the dynamics of information flow on social media. To satisfy such needs, in this study, we developed a new model, topic jerk detector, by expanding the existing burst detection method "topic dynamics." This model is ideal for identifying topic bursts; the main advantage of this method is that it is more accurately fitted to sudden bursts and detects the timing of the bursts of topics better than the existing method. It helps capture important topics that have rapidly risen to the top of the agenda in respect of time in the study of specific social issues. This expansion is achieved by expanding topic dynamics based on the physics analogy of velocity and jerk. Such a feature makes our method useful for monitoring tweets related to specific events. It is also expected to help understand phenomena we should be aware of in social media for proper information dissemination and management such as flaming phenomena.

To verify the effectiveness of the proposed method, we conducted three experiments. As detailed in Section 5.1, we applied the topic jerk detector to the dataset presented in Section 3 to detect the topics of interest to users at each period and their temporal variation. We succeeded in detecting some topics that attracted users' interest at each period. We also plotted the flow of users' interest in representative topics as a graph to observe their entire flow. In Section 5.2, we verified that the topic jerk detector effectively detects the rapid upward movement of a topic, compared to the existing method, topic dynamics. We concluded that the topic jerk detector detects the timing of the bursts more accurately. In Section 5.3, we interviewed two domain experts, and asked them to provide feedback on the method and its performance, to validate our method based on other aspects. They reported that our proposed method is effective, and can help to highlight popular topics that attract the attention of social media users. It is expected that the proposed method will facilitate the effective monitoring of the occurrence of topics in a stream of events in future disasters. There is also the possibility of an expert interpreting a series of events by linking the relevant topics at each point in time. However, there is some room for improvement, and we identified new research problems that shall be explored in our future research. The results of the experiments strongly verify the effectiveness of our proposed method. The method can be applied to other data, such as tweets on COVID-19, to clearly establish the kind of topics that strongly attract users' interests. Therefore, this study makes a significant contribution to the society.

**Author Contributions:** Conceptualization, H.N.; methodology, H.N.; formal analysis, H.N.; data curation, H.A.T.; discussion, T.H. and H.A.T.; writing–original draft preparation, H.N.; writing–review and editing, T.H. and H.A.T.; supervision, Y.O.; project administration, Y.O. All authors have read and agreed to the published version of the manuscript.

**Funding:** This work was partially supported by JSPS KAKENHI, Grant Numbers JP16H01836 and the Research on the Health Effects of Radiation initiative organized by the Ministry of the Environment, Japan.

**Acknowledgments:** We would like to express our deepest gratitude to Masaharu Tsubokura (Department of Radiation Protection, Soma Central Hospital, Soma, Fukushima, Japan) and Kazuko Uno (Louis Pasteur Center for Medical Research, Kyoto, Japan) for granting interviews as domain experts.

**Conflicts of Interest:** The authors declare no conflict of interest.

## Appendix A

**Table A1.** List of key phrases used for extraction of the data set for this study as well as for the previous study [18].

| Key Phrase | English Transition |
|---|---|
| 放射 | radio- or radia- |
| 被ばく, 被曝, 被爆 | exposure |
| 除染 | decontamination |
| 線量 | dose |
| ヨウ素 | iodine |
| セシウム | cesium |
| シーベルト, msv, μsv, usv, Sv, mSV, μSV, uSV | Sv, sievert |
| ベクレル, Bq | becquerel, Bq |
| ガンマ線, γ線 | gamma ray, γ-ray |
| 核種 | isotope |
| 甲状腺, 甲状線 | thyroid |
| チェルノブイリ | Chernobyl |
| 規制値 | regulation value |
| 基準値 | standard value |
| 学会 | academic society |
| 警戒区域 | no-entry zone |
| 避難区域 | evacuation zone |
| 産科婦人科 | obstetrics and gynecology |
| 周産期・新生児医 | perinatal and neonatal care |
| 日本疫 | nuclear medicine |
| 核医 | nuclear medicine |
| 電力中央 | central electric |
| 学術会議 | science council |
| 環境疫 | environmental epidemiology |
| 物理学会 | Physical Society |
| プルトニウム | plutonium |
| ストロンチウム | strontium |
| 暫定基準 | provisional standard |
| 暫定規制 | provisional regulation |
| 屋内退避 | sheltering |
| 金町浄水場 | Kanamachi Water Purification Plant |
| 出荷制限 | shipment restriction |
| 管理区域 | control area |
| 避難地域 | evacuation area |
| モニタリング | monitoring |
| スクリーニング | screening |
| ホットスポット | hot spot |
| 汚染 | contamination |
| (土 OR 食品 OR 水) AND 検査 | (soil OR food OR water) AND inspection |

**Table A1.** *Cont.*

| Key Phrase | English Transition |
|---|---|
| (がん OR ガン OR 癌) ANS リスク | cancer AND risk |
| 影響 AND (妊婦 OR 妊娠 OR 出産 OR 子ども OR 子供 OR こども OR 児) 母子避難 | effect AND (pregnant woman OR pregnancy OR childbirth OR child) mother and child evacuation |
| 避難弱者 | people having difficulty in evacuation |
| 自主避難 | voluntary evacuation |
| 避難関連死, 避難死 | death associated with evacuation |
| (安心 OR 安全 OR 不安 OR 食品 OR 野菜 OR 米 OR 牛肉 OR 産 OR 検査 OR 避難) AND (福島 OR ふくしま OR フクシマ) | (safety OR relief OR anxiety OR food OR vegetable OR rice OR beef OR product OR inspection OR evacuation) AND Fukushima |

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
