# Peer review of "Topic Jerk Detector: Detection of Tweet Bursts Related to the Fukushima Daiichi Nuclear Disaster"

_information, doi:10.3390/info11070368_

Round 1

Reviewer 1 Report

Dear Authors, 

Thank you for the interesting manuscript, but I have some comments:

Could you extend your abstract? Highline, the problem, the aim of your paper.

In my opinion, the text from the 12 till 22 lines is put not in write place, and please rethink this. 

I didn't find the aim of this paper...what is the problem...?

Maybe Table 1 you could replace in to annex section...

I assume that the table 3-5 have to be described and commented. 

In my opinion, the conclusions are conclusions, not summary and future work. Please rewrite it.

Reviewer 2 Report

The review of previous work is too focused on the Fukushima disaster. Methodology oriented related literature is a bit scarce. For instance, references related to anomaly detection in signals would be useful to assess the novelty of the method proposed.

In the last paragraph of the introduction, the authors claim “we expand the concept of topic dynamics as more fitting for burst data than the existing methods”. This claim has to be justified and details must be provided at this point in which sense this expansion is made.

Regarding the data, authors say that only 8% of tweets were considered. How this subsampling was designed? This can introduce a strong bias in the analysis and the interpretation and should be justified. As the authors could retrieve more complete data a check of representativeness should be made.

The authors make strong assumptions on topic modelling that should be justified with more references to frame the work they build upon.

Formula (1) is good, but the last term is confusing. Should not be df_{w,t}/N_t instead?

Section 5.1 is not well written and it makes very difficult to follow. It is not clear on which data the jerk detector is applied. A workflow picture would be useful.

The authors say “We listed the most frequently occurring words for each week”. It is not clear if this is based just on frequency of occurrences or it is based on the jerk indicator. It would be disappointing than tweets and words are selected based on simple statistics of number of retweets and frequency of words in tweets. This does not add value to topic identification.

The results really lack of data-based plots more than tables. The results are about dynamics so the authors should provide descriptions of dynamics rather than static tables.

Regarding parameters, a comparative result would be appreciated to properly characterize the method.

Section 5.2. lacks a total argumentation of the results. Why is this important? What the differences mean? What is the interpretation of the jerk detector

Section 5.3.: it is good that the authors try to validation with external experts, but two experts is a bit scarce sample for validation of the methodology and results. Questions should address false negatives, but also false positive detection of keywords. A discussion about the ranking itself would be useful.

In general the method is based on words more than topics. Topic is a broader and more general concept that requires experts or Machine Learning techniques to be classified. For instance a word can be relevant for characterize two different topics. The authors should discuss on this and present results according to this fact. This links with the experts’ concerns about the interpretability of the results. It also links to the problem of changes in topic that is a problem more complex paradigm than word occurrence dynamics.

A discussion on the time resolution required to measure differences in jerk should be provided. In twitter data volume of tweets can change in a daily resolution, so what information could a third derivative provide and add?

In the discussion the authors make some good reflections, but this should be part of the research as the results presented are rather short. The same happens with conclusions and future work. Further work to present a more developed research is encouraged at this point at the current manuscript is rather short regarding methodology and results.

In general the authors claim they propose a model to understand topic dynamics. Topic dynamics is something complex that requires a multidimensional approach. There is a huge conceptual gap between topic identification, topic dynamics and word statistics. Here, the authors seem to propose an indicator that is good to measure changes in words occurrence based on the jerk. However, the usability of this indicator is not fully explained and illustrated or justified.

Reviewer 3 Report

The topic is very important for these times because as it was the crisis in Fukushima in 2011 as well are these days of fear of spreading the coronavirus.

Social media has helped in many difficult situations because communication is the fundament of all things especially when is about solving a black swan.

In this paper, authors are emphasizing the need for understanding social media platforms that have been developing very fast. The social media current, have helped in 2011 with the Fukushima Daiichi nuclear power plant accident in Japan.

The proposal for the case study analysis, “topic jerk detector” is original and authors bring their own contribution. Authors applied the topic jerk detector to the database of historical tweets, for tracking trends and detecting discussion outbursts related to the Fukushima disaster. They used text mining and burst detection in user's discussions.

This ingenious method, the topic jerk detector, is suitable for identifying  sudden discussion bursts. We can also use this original method in detecting specific words for Coronavirus.

The paper is very good, the method and the tools for the statistical - math analysis is original and the contribution of the authors could be significant to the society.

Reviewer 4 Report

Suggestion

The results have shown that “Topic Jerk Detection” just has good effects on the detection of emergency topics,and it’s an improvement of “Topic Dynamics”.

   To see the effects of “Topic Jerk Detection”,the authors could add  a model that also deals with sudden outbursts of discussions and compare it with the effect of reaction topic jerk detector?

Spelling:

Line 15: “in Twitter” change to “on Twitter”

Line 23: “accident” change to “accidents”;

“the radiation” change to “radiation”

Line 33: “information” change to “the information”

Line 218: “the answers of the two domains”

change to “the answers to the two domains”

Line 219: “the type of the incident”

change to “the type of incident”

Line 228: “the social media users”

change to “social media users”

Line 235: “accomplished different objectives”

 change to “different accomplished objectives”

Line 239: “a certain topic bursts”

change to “a certain topic burst”

 or “certain topic bursts”

Round 2

Reviewer 1 Report

Thank you for your comments and improvement.

Author Response

Thank you for your comments and improvement.

→ We appreciate your beneficial comments, too.

Reviewer 2 Report

Thanks for the authors for addressing all points of the review. For future interactions, a version with marked changes would really help tracking the improvement.

The introduction still lacks a more concrete justification of why social media are important in disaster management. The introduction goes from very generic claims about Twitter to the method without providing an illustrative perspective. Some studies have addressed this question, for instance, https://www.mdpi.com/2071-1050/12/10/4246 in which the authors use Twitter in several ways to improve disaster understanding and management. A motivation statement would be useful to assess the impact of the proposed work into real-world application for disasters.

The number of references seems still scarce. The literature regarding information spreading can be mentioned. For instance, Barabassi’s lab work should be mentioned along with similar works.

The authors keep saying that they detect topics rather than keywords. It is not accurate to identify topics with keywords as topic is a semantic-level description of text that involves the meaning of several words and their use. In case they really do just keyword analysis, it should be said that they assign topic and keyword the same concept within this work, which is obviously not good, but at least it should be said. Some references in topic analysis using Machine Learning should be provided for clarification.

The new section 2.2. fails to provide a comprehensive view of the topic burst detection landscape. This section is a starting point, but it should be improved.

At the end of section 2.2. claim: In this study, we developed a method termed “topic jerk detector,” an expansion of topic dynamics; the topic jerk detector is more effective for detecting “bursting” than topic dynamics, especially in terms of figuring out the specific topics the public has a great deal of interest in. Due to this uniqueness, it is expected that the proposed method can recognize hot words in each term among users, and track the transition of topics more effectively.” This is the main claim that should be said in the introduction as well. From here two questions arise that have to be solved:

  • Can the topic jerk detector detect hidden topics that cannot be detected with the “topic dynamics”?

  • In terms of temporal evolution, does the topic jerk detector detect “topics” earlier? so this could have implications in the mechanisms of response.

It is completely necessary to address these points throughout the document.

The data used is filtered by keywords, what is the impact of this filtering in the methodology?  Do the detected topics fall under these keywords? How would it work to predict disasters in a real-world scenario?

Section 5.1. Thanks to the authors for including new figures. Figure 3 is supposed to show people in Tables 2, 3, but these tables do not contain user information. The results of Figure 3 are nowhere introduced and explained properly. How the jerk detector is associated to people?

It is not clear why authors select the 100 most “relevant” tweets. Do they only apply the analysis to these tweets? It is confusing. Does the period refer to years? It should be explicitly said what are the intervals.

Section 5.1 should be oriented to respond if it is possible to discover hidden topics that become important. How this could be scaled up? How can be used to automatically detect the topic jerks? This is relevant for usability of the method.

The authors responded “The new figures 4 and 5 describe the dynamics of the main topics”. However, figures 4 and 5 are similar to the ones provided in the previous version. This should be clarified.

Section 5.2 should be oriented to respond the other question about the timing of the detection. What is the gain in the detection timing? Hours, minutes, days? This would be the main value for operational procedures using the topic jerk detector. It should provide more quantitative evidences of the temporal gain. This is not clear in the current figures.

Authors said they “conducted some additional surveys”, but this is not presented in the new version of section 5.3, please clarify.

Last two sections have been improved, but the authors fail to respond the two questions posed in this review about the automated discovery and the temporal gain in the detection in quantitative terms.

Round 3

Reviewer 2 Report

Thanks for the authors for marking the changes in the new version and addressing the comments.

The abstract should rely on a specific result statement as the phrase “Thanks for the authors for marking the changes in the new version and addressing the comments.” is too generic and vague.

The authors claim that they can provide an explanation for flaming in the social media. The authors introduce “flaming” explanation as a new feature/impact of the method. More concretely the authors claim “For efficient information sharing, it is necessary to understand why the flaming occurred to prevent the recurrence.” However, this is repeated several times along the manuscript without an explanation or justification.

Regarding the application of the method to “real-world” cases, the authors should explain better how to make an analysis real-time vs analysis offline. Also, the filtering of posts already narrows down the set of potential keywords that can be detected with the method. Thus, the scope of the method to detect topics of interest gets very limited. The completeness of topics related to a disaster should be discussed.

The authors say that the method is not intended to find hidden topics nor earlier detection of topics. Thus, it is not clear the advantage of the method. For instance, how it differs the ranking provided by the jerk detection to other detection methods or a simple frequency analysis of keywords? There is not a clear epistemological advantage in this method. This kind of comparative analysis is critical to show the value proposal of this work.

It still not clear how this method is more effective to detect hot topics. Which metrics are used to support this claim? the mere conceptual difference is not sufficient to claim this, even more when they authors do not confirm that the advantage is a more timely detection provided by the jerk-level measurement.

Authors also claim “It makes it possible to monitor the occurrence of topics in a stream of events, especially topics that greatly interest the public.”. Is it any special metric to assess the interest of the public? In which sense the ranking allows inferring the interest of the topics? These claims should be supported with evidence and comparative studies and ground-truth data. Without ground-truth data is very difficult to justify the level of interest of topics which is a qualitative assessment that cannot be reduced to the occurrence of keywords. Then the authors talk about influential tweets, there is no justification to go from occurrence to influence. These terms are used imprecisely when they are clearly differentiated conceptual aspects that should not be mixed.

It is necessary that the authors show that the ranking of “topics” provided by this method is different and more relevant to understand the disaster than other methods.
